# Robust processing of airborne laser scans to plant area density profiles

Johan Arnqvist[1], Julia Freier[2], and Ebba Dellwik[3]

[1]Johan Arnqvist, Uppsala University, department of Earth Sciences, Uppsala, Sweden
[2]Fraunhofer Institute for Energy Economics and Energy System Technology, Kassel, Germany
[3]Technical University of Denmark, Roskilde, Denmark

**Correspondence:** Johan Arnqvist (johan.arnqvist@geo.uu.se)

**Abstract.** We present a new algorithm for the estimation of plant area density (PAD) profiles and plant area index (PAI) for forested areas based on data from airborne lidar.

The new element in the algorithm is to scale and average returned lidar intensities for each lidar pulse, whereas other methods either do not use the intensity information at all, only use average intensity values or do not scale the intensity information, which can cause problems for heterogeneous vegetation. We compare the performance of the new and three previously published algorithms over two contrasting types of forest: a boreal coniferous forest with a relatively open structure and a dense beech forest. For the beech forest site, both summer (full leaf) and winter (bare trees) scans are analyzed, thereby testing the algorithm over a wide spectrum of PAIs.

Whereas all tested algorithms give qualitatively similar results, absolute differences are large (up to 400 % for the average PAI at one site). A comparison with ground based estimates shows that the new algorithm performs well for the tested sites. Specific weak points for estimation of PAD from airborne lidar data are addressed; the influence of ground reflections and the effect of small-scale heterogeneity, and we show how the effect of these points is reduced in the new algorithm, by combining benefits of earlier algorithms. We further show that low-resolution gridding of PAD will lead to a negative bias in the resulting estimate according to Jensen's inequality for convex functions, and that the severity of this bias is method-dependent. As a result, PAI magnitude as well as heterogeneity scales should be carefully considered when setting the resolution for PAD gridding of airborne lidar scans.

## 1 Introduction

Plant area is a key parameter for quantification of air-vegetation exchange of momentum, latent and sensible heat as well as carbon dioxide. Its vertical distribution, the plant area density (PAD), describes the distribution of plant elements from ground to canopy top. The PAD profile is a key parameter in high-accuracy atmospheric models for the estimation of fluxes between the atmosphere and canopies (Williams et al., 1996; Sogachev et al., 2002; Patton et al., 2016; Smallman et al., 2013). PAD profiles have also been introduced into wind modelling of heterogeneous forests in the context of understanding how local inhomogeneities affect the wind field above the crowns (Boudreault et al., 2015; Ivanell et al., 2018). Since wind turbines are very tall, the wind approaching the turbine is affected by surface conditions from far away. Therefore, large areas of consistent

high-quality PAD profiles are attractive as model input. Likewise, atmosphere-biosphere interactions for weather and climate modelling require PAD information from large areas. With the advent of airborne lidar scans (ALS), such a product is possible to achieve.

ALS scans, made with a density of 5-10 laser shots over each square meter of ground are now performed routinely on a country level scale. Besides information on where the laser beam from the airplane was reflected in $x$, $y$ and $z$ coordinates down to centimeter precision, each point is associated with several other attributes. These normally include scanning angle, number of returns from a pulse, return rank in the pulse and intensity of the return. In this paper, we attempt to use more of this information compared to previously published methods (Solberg et al., 2006; Morsdorf et al., 2006; Richardson et al., 2009; Boudreault et al., 2015; Almeida et al., 2019; Hopkinson and Chasmer, 2009; Vincent et al., 2017) with the aim of producing a more consistent and less-biased estimate of PAD.

The previously established methods for calculating the PAD profile, and hence also the PAI, from ALS scans, were based on approximations of the Beer-Lambert law, first introduced in the field of vegetation density estimation by Monsi and Saeki in 1953 (see Monsi and Saeki (2005) for a translated version). The results have been promising for PAI (Solberg et al., 2006; Morsdorf et al., 2006; Richardson et al., 2009; Hopkinson and Chasmer, 2009) and even PAD (Boudreault et al., 2015; Vincent et al., 2017; Almeida et al., 2019) calculations by ALS, when compared with various ground based methods. The potential is underlined by relative agreement with ground based methods made with very different techniques as well as the obvious advantage of producing results with great spatial coverage. Below, we recall the Beer-Lambert law and its adaptation for vegetative canopies, and then describe three reference models, published by Hopkinson and Chasmer (2009), Boudreault et al. (2015) and Almeida et al. (2019). We then introduce the new method. The rest of the paper follows standard formats.

## 2 Background

### 2.1 Beer-Lambert law and its adaptation for vegetative canopies

The Beer-Lambert law expresses the light level at a given height as a function of effective canopy density in form of PAD following:

$$Q(z) = Q_0 e^{\frac{-\mu}{\cos\theta} \int_z^{h_c} PAD(z) \mathrm{d}z}, \tag{1}$$

where $Q$ is the amount of radiation that penetrates the canopy to height $z$, $Q_0$ is the incoming radiation, $\mu$ is the extinction coefficient, $\theta$ is the zenith angle of the incoming radiation and $h_c$ is the height of the canopy. The value of $\mu$ depends on how the canopy elements are oriented in space, and the values for common types of trees have been reported in the range of 0.28 to 0.58, see Bréda (2003) for a detailed investigation. By re-arranging Eq. 1, PAD can be isolated and light-level observations can provide a basis for estimations of vegetation density.

In contrast to ground based optical methods, measurements from ALS are based on the *reflected radiation R*. Whereas it is clear from the Eq. 1, that the radiation level above the canopy $Q_0$ is of high importance for the determination of the determination of PAD and PAI, the use of the reflected radiation turns the problem on its head. To determine the value of the PAI based

on ALS data, ground reflections are essential and both for the PAI and the PAD the magnitude of the incoming radiation is not important provided it is sufficiently strong to produce detectable backscatter. There are several other fundamental differences between the ground based optical methods for determining PAI and the lidar-based methods; (1) Whereas optical methods are typically based on observed radiation of natural light, ALS normally use infra red light, (2) the primary information utilized in ALS-based methods is the exact location describing where the radiation is reflected (e.g. Morsdorf et al. (2006); Solberg et al. (2006)) and (3) contrary to the homogeneously lit canopies used to determine plant area using ground based optical methods (Monsi and Saeki, 2005; Bréda, 2003; Yan et al., 2019), the ALS data are based on discrete lidar pulses, which have a finite diameter of around 0.1 - 1 m at canopy top (*e.g.,* Hopkinson and Chasmer, 2009; Popescu et al., 2011; Almeida et al., 2019). The high sampling rate of the ALS scanners allow for the detection of multiple reflections from one single emitted pulse, with data sets ranging from one to several returns per pulse, all the way up to the full wave form, depending on the type of scanner. Regardless of these fundamental differences, the Beer-Lambert law has been used also for the determination of PAD and PAI using ALS data using the following formulation:

$$\frac{\int_0^z R(z)\mathrm{d}z}{\int_0^{h_c} R(z)\mathrm{d}z} = e^{\frac{-\mu}{\cos\theta} \int_z^{h_c} PAD(z)\mathrm{d}z}, \tag{2}$$

where $R$ is the reflected radiation at height $z$, and integration on the right hand side all the way from the forest floor gives the PAI.

The method for determining PAI and PAD are based on a grouping of the exact coordinates where the pulses have reflected on a canopy element, into distinct vertical layers $1, 2, ..., k, ..., k_{top} - 1, k_{top}$, where 1 is the layer closest to the ground and $k_{top}$ is the highest layer containing plant area density. The effective PAD (assuming constant extinction coefficient, which implies locally homogeneous distribution of canopy elements) between layer $z_k$ and $z_{k+1}$ can be found by isolation in Eq. 2:

$$\overline{PAD}\Delta z = -\frac{\cos\theta_l}{\mu} \ln\left(\frac{\sum_{i=1}^k R_i}{\sum_{i=1}^{k+1} R_i}\right) \tag{3}$$

where the overbar indicates average, $\Delta z = z_{k+1} - z_k$ and $\theta_l$ is the zenith angle of the lidar beam.

Despite the significant differences behind airborne and ground based estimates of PAI, comparisons have shown promising agreement in both absolute levels as well as spatial correlations (Solberg et al., 2006; Morsdorf et al., 2006; Richardson et al., 2009; Solberg et al., 2009). A straightforward interpretation of the correlation was put forward by Solberg et al. (2009), who concluded that the canopy gap fraction was strongly correlated to the probability of ALS beam penetration below the canopy, and hence, strongly linked to PAI. Boudreault et al. (2015) and Almeida et al. (2019) also compared PAD profiles to observations from reference sensors, with similarly promising results.

Whereas earlier ALS studies used ground based estimates of PAD and PAI as validation, it is generally acknowledged that also ground based methods carry uncertainty, for instance regarding sky exposure level and difficulties in reproducing footprint, points often brought up in the discussion sections of earlier work (Solberg et al., 2006; Morsdorf et al., 2006; Richardson et al., 2009; Solberg et al., 2009; Vincent et al., 2017). For the datasets presented here, ground based reference data are included for

two out of the three investigated cases. However, the main focus of the study is instead the comparison between different ALS methods.

## 2.2 ALS Reference methods

Two different interpretations of the reflected radiation $R$ in Eq 3 have been applied to ALS data; one based on measured *intensities* of the reflected pulses (Hopkinson and Chasmer, 2009) and one based on the relative probability of detecting a reflection (Solberg et al., 2006; Morsdorf et al., 2006; Richardson et al., 2009; Boudreault et al., 2015; Almeida et al., 2019). The latter method can be subcategorized further based on how many reflections (or returns) from each lidar pulse are considered in the PAD and PAI evaluation. Hence, in total, we use the following three reference methods:

– **Intensity ratio, IR**. This method makes use of the intensity $I$ of the recorded reflection following Hopkinson and Chasmer (2009). Over a volume $\Delta A \Delta z$, the right-hand side of Eq. 3 is calculated from:

$$\frac{\sum_{i=1}^{k} R_i}{\sum_{i=1}^{k+1} R_i} = \frac{\sum_{i=1}^{k} I_i}{\sum_{i=1}^{k+1} I_i} \tag{4}$$

It is assumed that the reflected intensity from a vertical section of forest is dependent on vegetation density from that section. An important assumption for IR method is that the albedo $\alpha = R/Q$ is the same for ground and vegetation.

– **First returns ratio, FR**. This method has been used in Solberg et al. (2006, 2009); Morsdorf et al. (2006); Richardson et al. (2009); Boudreault et al. (2015); Almeida et al. (2019) and here the amount of reflected radiation $R$ is rather interpreted as the number of reflections from a distinct layer of the canopy. The method is limited to using only the first returns ($r_1$) for each pulse and it is assumed that the probability of pulse penetration into the canopy decreases exponentially with height:

$$\frac{\sum_{i=1}^{k} R_i}{\sum_{i=1}^{k+1} R_i} = \frac{\sum_{i=1}^{k} r_{1_i}}{\sum_{i=1}^{k+1} r_{1_i}} \tag{5}$$

– **All returns ratio, AR**. Assuming the intercepted radiation is related to the total number of returns from a volume $\Delta A \Delta z$, no matter the return number in the pulse. (Almeida et al. (2019), supplementary material).

$$\frac{\sum_{i=1}^{k} R_i}{\sum_{i=1}^{k+1} R_i} = \frac{\sum_{i=1}^{k} r_i}{\sum_{i=1}^{k+1} r_i} \tag{6}$$

The different methods are associated with different uncertainties and advantages. Starting from below, Almeida et al. (2019) noted that the AR method has a poor theoretical foundation in that the total number of emitted pulses above the canopy are not same as the number of reflections accounted for, and pointed to the need for further algorithm improvement. They also noted a poorer agreement with reference data than when using the FR method, which gives an unbiased probability of interception. However, the FR method is associated with other documented issues, especially for dense canopies (Blair and Hofton, 1999; Richardson et al., 2009; Freier, 2017), where the diameter of the laser beam (footprint size) may exceed a typical size gap in

the canopy. Since the FR method is based on the probability that the beam passes through the canopy top with no obstruction, the method becomes sensitive to the ratio of the lidar footprint and the typical gap size in the canopy, and a high site-to-site variability is expected. A further disadvantage of the FR method is that a large part of the dataset is disregarded, since secondary reflections are not utilized.

The IR method in many ways circumvents the above issues; all reflections are used, and further, since the method is formulated in terms of intensity rather than relative probability, the framework is closely related to the original Eq 1. Another obvious advantage is that by using the intensity information for the recorded reflection, information on the size of the reflecting object can be extracted; a reflection on a thin bare twig will result in a low-intensity reflection, whereas higher intensity reflections can be expected from a fully leaved branch. Hopkinson and Chasmer (2009) showed improvements using the intensity information

compared to using only the first returns (based on correlation with gap fractions from hemispheric photos) and concluded that an intensity based method is less error prone due to the implicit information about the size of the return object that is carried in the intensity value. A disadvantage of the IR method is that it relies on the assumption of identical albedo for ground and vegetation. Wagner et al. (2008) for instance, reported on difference in reflective properties between non-vegetation and vegetation. This is particularly problematic for less dense forests were many of the returns come from the ground. In the IR method, the

intensity value of first return ground reflections partly determine the PAD of the upper canopy, so PAD and PAI estimates from the method are sensitive to the properties of the ground.

    Thus, a method that combines the respective strengths of the IR and FR methods has potential to be less sensitive to parameters that affect beam width, such as flight altitude and instrument type, and thereby be more suited for PAI and PAD estimates from ALS.

In order to ensure a comparison where the treatment of the reflections are consistent, the three references methods have been slightly modified relative to their original formulations. For the AR method, we use the formulation presented in the supplementary material by Almeida (2019), with the modification of taking the scanning angle into account in the same way as done in the FR method by Boudreault et al. (2015). For the IR method, we follow the one way formulation by Hopkinson and Chasmer (2009), by which the two-way transfer of the pulse through the crown is disregarded. The reason for only using one-

way attenuation is that the scattering from canopy elements and ground does not come from continuous attenuation, but rather distinct reflection on objects. That, in combination with the fact that changes in $\theta_l$ (position of the airplane) are negligible under the short time it takes for the light pulse to return to the sensor, leads to the conclusion that if the path was free of obstacles on the way down, it should also be free on the way up.

## 3   Methods

### 3.1   New method

The new method will be referred to as SR, Scaled Ratio. Using this method, all reflections from each emitted pulse are identified in a preprocessing step. Once the reflections are identified, their impacts are scaled according to their intensities in a similar

way as in the IR method. Mathematically, this individual beam scaling can be expressed as:

$$r_s = I_{i_r} \Big/ \sum_{i_r=1}^{N_r} I_{i_r}, \tag{7}$$

where $I_{i_r}$ is the intensity of the $i_r$:th return in the pulse, and the scale is found by summation from the first return in the pulse, to the number of returns that the pulse contain ($N_r$). After the intensity scaling the PAD is calculated in the same way as with the other methods; (1) by estimating the ratio of incoming to outgoing radiation following

$$\frac{\sum_{i=1}^{k} R_i}{\sum_{i=1}^{k+1} R_i} = \frac{\sum_{i=1}^{k} r_{s_i}}{\sum_{i=1}^{k+1} r_{s_i}}, \tag{8}$$

and (2) by calculating PAD using Eq. 3. In this way, for all pulses with only one return ($N_r = 1$), the scaling weight is 1, whereas the weight for all other pulses are scaled by their respective backscatter intensity. Hence, for the special case of a data set with only first returns, the SR method reduces to the FR method and for the special case of only one pulse over the binning area $A$, the SR method reduces to the IR method. For the majority of modern ALS datasets, the scanning density is high, and the SR method has the potential to combine the benefits of the FR and SR methods, while minimizing their drawbacks regarding too high influence from difference in ground/vegetation albedo as well as footprint size relative to canopy gap size.

The SR method requires that the backscatter intensity of each return can be connected to the total backscatter intensity of the emitted pulse which the return belongs to. In order to test this assumption, we estimated what fraction of the reflections in the datasets fulfilled the criterion that with $i_r = N_r$ along with $N_r - 1$ preceding points satisfy $\Pi_{i_r=1}^{N_r} i_r = N_r!$. For the data sets that has been part of this study, the number of points satisfying the above criterion was more than 99 % of all the investigated data sets.

Matlab and Python codes for the preprocessing step, the scaling and the calculation of PAD are shared via the link the Supplementary material. The routine requires .las data files, which are the global standard for storing ALS data (ASPRS, 2013), as inputs, and outputs terrain height, PAI and PAD at a horizontal and vertical resolution set by the user. An early version of the SR algorithm was tested in Freier (2017).

The SR method uses all of the standard recorded attributes of the reflections (Table 1) in the .las format, whereas the reference methods (in addition to zenith angle $\theta$) use either only the position and classification (AR), position, classification and return number (FR), or position, classification and intensity (IR).

In the comparison with reference methods we follow Solberg et al. (2006); Morsdorf et al. (2006); Richardson et al. (2009); Boudreault et al. (2015) in assuming a spherical distribution of the reflecting surfaces of vegetation, corresponding to a value of $\mu = 0.5$, equivalent of approximating that the effective forest density is the same looking from above, from the side, or from any other direction of the forest. While choosing a value of $\mu$ fitting for the vegetation of interest improves the PAI estimates (Yan et al., 2019), the focus of this study is the application of Beer-Lambert law itself, and any change in $\mu$ would be common for the investigated methods, and therefore the value $\mu = 0.5$ has been kept throughout the study. To speed up the calculation, instead of using pulse specific $\theta_l$, we have used the average absolute angle within the grid box.

Recently, Vincent et al. (2017) presented a method in which intensity information is used to determine averaging weights specific to return number, similar to the method presented here. However, the method proposed by Vincent et al. (2017) use

**Table 1.** Attributes used to calculate the PAD from ALS with four different methods. The attributes are usually available with data in the .las format

| | Scaled return ratio (SR) | Intensity ratio (IR) | First return ratio (FR) | All returns ratio (AR) |
|---|:---:|:---:|:---:|:---:|
| Position, $X, Y, Z$ | × | × | × | × |
| Intensity, $I$ | × | × | | |
| Return number, $i_r$ | × | | × | |
| Number of Returns (per pulse), $N_R$ | × | | | |
| Classification (ground identification) | × | × | × | × |
| Scanning angle, $\theta_l$ | × | × | × | × |

**Table 2.** Characteristics of the three ALS data sets used in the study. $^\star$ Lantmäteriet (2016). $^\dagger$ Personal communication with data provider.

| Scan | 1:st returns/m$^2$ | Footprint diameter [m] | % pulses with $N_r = 1$ | % pulses with $N_r = 2$ | % pulses with $N_r > 2$ |
|---|---|---|---|---|---|
| Norunda | 0.9 | 0.5-0.8$^\star$ | 73 | 23 | 3 |
| Falster summer | 32.5 | 0.1$^\dagger$ | 53 | 40 | 8 |
| Falster winter | 8.3 | 0.4$^\dagger$ | 47 | 30 | 23 |

the intensity information averaged over the area of interest while the method proposed in this paper uses individual intensity data for each pulse. If the individual intensities are not available in the data set, or the data set is not ordered in consecutive returns, using the average data might be the only option. For a discussion on the implications of using average intensities as scales, please see Section 5.1.

## 3.2 Sites

The methods were tested on three different data sets from two different sites. One is a dense deciduous (beech) forest, scanned both with and without leaves, and the other one is a medium dense coniferous (mainly pine) forest. The data sets were chosen based on that they cover two different and commonly occurring types of forests as well as the availability of both ALS data and ground based PAI estimates. Table 2 shows a summary of scan characteristics for the three different data sets. The width of the laser beam at ground differ from around 10 cm in the beech summer scan up to 80 cm in the coniferous forest, due to different instruments and flight altitudes. The scanning density also differ between sets (between less than 1 and more than 32 returns per m$^2$), providing contrasting conditions for testing the ALS to PAD methods.

**Dense beech forest: Tromnaes**

This approximately 80 year old forest of predominantly European beech (*Fagus sylvatica* L.) is located near the coast on the island Falster in Denmark. During the leaf-on summer period the canopy is very dense, whereas the winter time tree structure is relatively open as seen in Fig. 1. In order to accurately estimate the canopy height and density for the purpose of wind flow modelling, the forest was scanned during the summer time in 2013. The site was scanned again in a national survey in 2014

during the leaf-off winter period. In connection with a field campaign of wind observations upwind and downwind of the forest edge, the forest density was further measured using two LAI-2000 PCAs (LI-COR, Lincoln, NE, USA). The PAI was found to be approximately 1 and 6 $m^2\,m^{-2}$ during leaf-off and leaf-on conditions, respectively. These measurements were performed using two instruments; one was placed well within the forest and the other, which was used for reference measurement, was placed well outside the forest. More information on the site can be found in Dellwik et al. (2014).

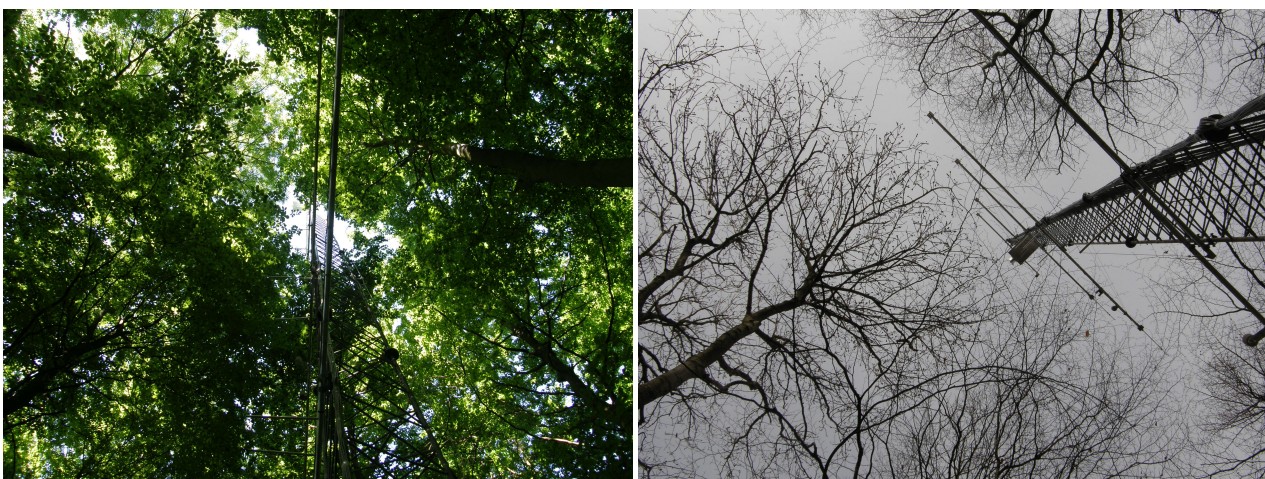

**Figure 1.** Photos taken at the position of the beech forest mast at the Falster site (Dellwik et al., 2014) illustrating the large difference between summer (left) and winter (right) canopies.

From the point cloud of the two scans, we focus our analysis on a 300 m $\times$ 300 m tile which has the center coordinate at 54.7638°N, 12.0396°E. The data sets will be referred to as *Falster winter* and *Falster summer*.

### Norunda

The Norunda site is a research site under the ICOS, Integrated Carbon Observation System, infrastructure, located in the south east Sweden (60.0833°N, 17.4833°E). The forest cover is predominately Norway spruce (Picea abies) and Scots pine (Pinus sylvestris). A minor fraction (15 %) of deciduous trees, mainly birch (Betula sp.) is also present. The forest at the site is marked by its many clearings, characteristic of Swedish forest management. As part of the ICOS project, the forest density at the site has been analyzed with digital hemispheric photometry according to the ICOS protocol for ancillary vegetation measurements (Gielen et al., 2018). The site has also been scanned with airborne laser as part of a national scan over Sweden (Lantmäteriet, 2016). The scan was made in November 2010 and the coverage is generally more than 0.5 ground reflections per $m^2$, but sometimes as low as around 5 ground reflections per 10 m $\times$ 10 m square in the most dense parts of the forest. The data set will be referred to as *Norunda*.

Ground based estimates of the forest were made in November 2018. Despite the difference of eight years between the ground based measurements and the airborne scan, the two data sets are expected to be comparable, since the age of the particular forest stand was more than hundred years and thus the growth rate was very low.

### 3.3 Evaluation parameters

**Spatial differences**

Spatial differences are demonstrated by showing differences of PAI from the new method (SR) relative to the reference methods (IR, FR and AR) for larger areas using a resolution of $10\,\mathrm{m} \times 10\,\mathrm{m}$ for the three used ALS scans. Calculation followed the same procedure for all methods, with the only distinction being the use of Eq. 4, 5, 6 or 8 as a model for the extinction.

**Comparison to ground based methods**

To provide an independent comparison, PAI from ALS are compared to ground-based estimates. To mimic the footprint of the ground-based methods, reflections within a radius of one tree height from the centre of the ground based measurement were used. This yielded a circular binning area, as opposed to the square used in all other comparisons.

**Sensitivity to ground albedo**

In order to investigate the sensitivity to ground albedo, a test was constructed where the intensity values of returns classified as ground were manipulated. Ground intensity values were increased by 10 % after which PAI was calculated and compared to PAI derived from the original data set. The same test was done by instead decreasing by 10 %. The manipulation only affects SR and IR, as FR and AR excludes intensity information (Table 1). SR and IR showed only minor differences in sensitivity between a 10 % decrease and a 10 % increase in ground intensities, so results are presented as sensitivity to a 10 % ground intensity alteration.

**Sensitivity to grid size**

All four ALS methods rely on the Beer-Lambert law which assumes homogeneous spatial distribution of the scattering elements. The assumption is generally violated in forests, which have both vertically and horizontally inhomogeneous distribution of scattering elements. With vertical binning in thin planes, the effect of vertical inhomogeneity is minimized, but heterogeneity in the horizontal is still present, and results in a grid size sensitivity of the ALS to PAD methods. The reason for this can be illustrated by studying a single layer PAI calculation for which the Beer-Lambert law simplifies to

$$\mathrm{PAI} = -\frac{\cos\theta_l}{\mu} \ln\left(\frac{\sum_{\mathrm{ground}} R_i}{\sum_{\mathrm{all}} R_i}\right) \tag{9}$$

Calculating the PAI of an area by reducing the horizontal resolution so that there is only one single cell for the whole area is then proportional to $-\ln\left(\overline{\sum_{\mathrm{ground}} R_i / \sum_{\mathrm{all}} R_i}\right)$, where the vertical bar represents area average. Calculating the PAI for the

whole area by taking the average of PAI from a higher resolution is instead proportional to $\overline{-\ln\left(\sum_{\mathrm{ground}} R_i / \sum_{\mathrm{all}} R_i\right)}$. Because of the curvature of the convex logarithm function, Jensen's inequality states that $-\ln\left(\sum_{\mathrm{ground}} R_i / \sum_{\mathrm{all}} R_i\right) > -\ln\left(\overline{\sum_{\mathrm{ground}} R_i / \sum_{\mathrm{all}} R_i}\right)$ which means that sub grid size variations in the ratio of canopy to ground returns leads to lower PAI estimates for larger grid size. The sensitivity of the routines to grid size was investigated by simply calculating PAD and

5    PAI for the three data sets with different horizontal resolution, ranging from $10\,\mathrm{m} \times 10\,\mathrm{m}$ to $100\,\mathrm{m} \times 100\,\mathrm{m}$. Before calculating PAI and PAD the terrain height data was flattened in the vertical direction by subtracting the $10\,\mathrm{m} \times 10\,\mathrm{m}$ ground height in order to avoid effects coming from terrain elevation changes within the grid cell.

## 4   Results

### 4.1   Spatial differences and comparison to ground-based methods

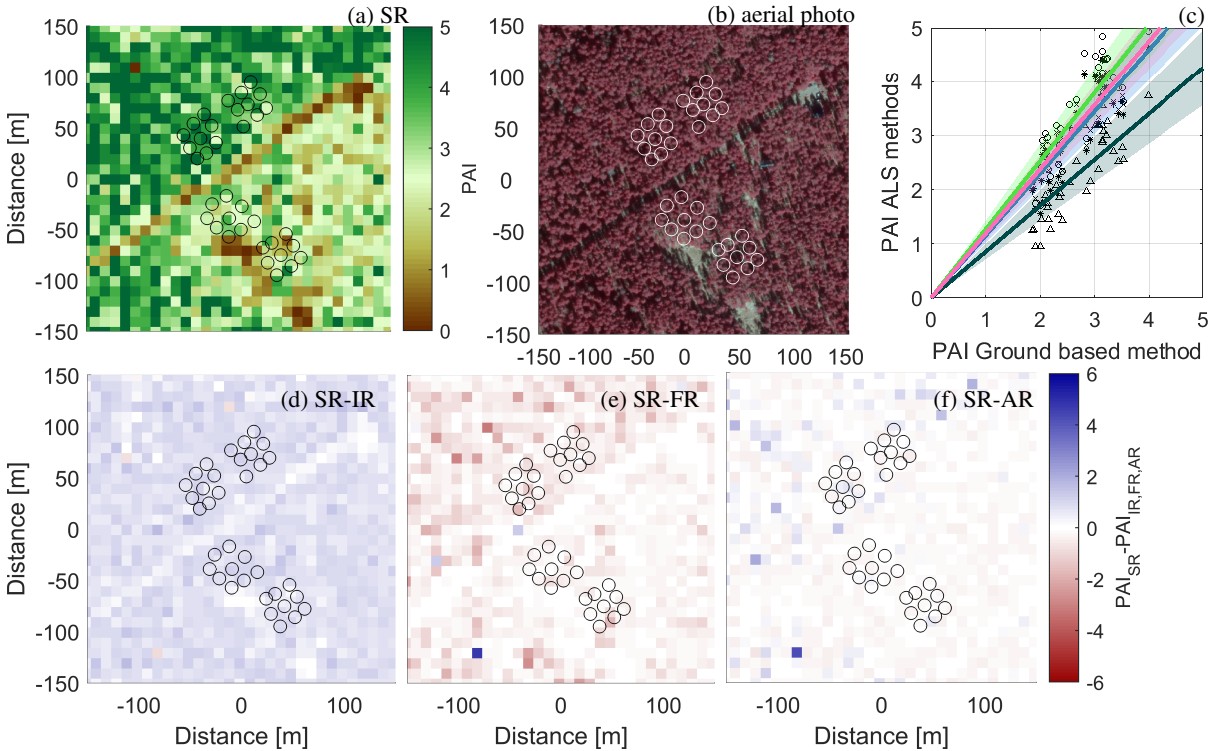

**Figure 2.** PAI evaluation at the Norunda site in a $10\,\mathrm{m} \times 10\,\mathrm{m}$ grid : $a$) the SR method, $b$) an aerial photo over the site, where the circles represent the placement of the ground-based observations, $c$) scatterplot of the hemispheric photo estimates and the ALS methods. The lines show linear regression forced through zero with shading indicating 95 % confidence level (1.96 times the standard error). SR is represented by blue and stars, IR is represented by dark green and triangles, FR is represented by bright green and circles and AR is represented by pink and crosses. $d$)-$f$) the difference between SR and the reference methods IR, FR and AR respectively.

Maps of the calculated PAI can be seen in Fig. 2, Fig. 3 and 4 for the Norunda and Falster sites, respectively. PAI magnitudes are shown for the SR method, and for IR, FR and AR, the difference to the SR method is shown. It is immediately clear that forest characterization by ALS has a big advantage compared to ground-based estimates due to the full horizontal coverage. In Fig. 2b, details of a road and clearings at the site shown in the aerial photo are clearly visible also in the PAI estimates. In

terms of PAI magnitudes, FR gives the highest values for all three data sets (Fig. 2e, 3e, 4c), whereas the IR method shows the lowest values (Fig. 2d, 3d, 4b). Average PAI values over the area at the $10\,\mathrm{m} \times 10\,\mathrm{m}$ resolution are given in the first column of Table 3, showing that for Norunda, the differences between the IR and SR methods is approximately 25%, whereas it is less than or equal to 12% for the AR and FR methods, respectively. In addition to these differences, larger local differences up to approximately 100% are visible in Fig. 2 (d-f). A comparison to the ground-based PAI observations is presented in Fig. 2

(c). Whereas the IR method shows a systematic underestimation relative to the ground-based observations, all other methods show an overestimation. These systematic differences could likely be removed by calibrating the extinction coefficient $\mu$, as was done in *e.g.* Solberg et al. (2009), but it is clear that such calibration would need to be method-specific, and it is therefore not pursued further here. Another reason for an imperfect match is connected with the unknown footprint of the ground-based method, see Section 4.3 below.

Whereas the results from SR, FR and AR were relatively similar in Norunda, a larger difference is seen for these methods for the summer scan at the dense beech forest at the Falster site (Fig. 3). The SR method indicates a large variability over the tile (Fig. 3 (a)), with PAI exceeding $10\,\mathrm{m^2 m^{-2}}$ in the lower right and upper left corner. A close look at the aerial photo (Fig. 3 (b)) also indicates larger gaps in the forest in the lower left of the image, which could explain the relatively lower PAI in this area. Whereas the observed spatial variability using the SR method is similar for the IR method (d), it is strongly increased for

the FR method (e) and decreased for the AR method (f). The very high PAI values using the FR method (e) can be explained by few pulses penetrating the dense upper canopy, in line with earlier observations on gap fraction versus beam footprint. The AR results are generally lower in magnitude than SR, whereas the FR are of higher magnitude. Averages over the whole tile (first column, Table 3) reveal mean differences up to 30% relative to the SR method.

The comparison between the ALS and ground-based observations (Fig. 3 (c)) reveals a high degree of scatter and large

standard errors. In addition to the potential reasons for the difference given above for the Norunda site, there is more uncertainty regarding the exact coordinates of the ground-based observations. In general, the ground-based observations seem to be saturated at around $6\,\mathrm{m^2 m^{-2}}$, whereas the ALS data allows for a larger range.

Fig. 4 shows the spatial variability over the same area as in Fig. 3, but for the winter scan of the site. Overall the SR results indicate low PAI (a), except for a small area to the right of the clearing, where values above $3\,\mathrm{m^2 m^{-2}}$ are visible, corresponding

to the location of a few coniferous trees. The IR method shows similar results to the SR (b), whereas much larger differences are observed for the FR and AR methods (c) and (d). The mean value over the whole tile (first column, Table 3 indicates differences up to approximately 300%. The reason for this is examined in detail in the next section, but as can be seen from Fig. 1, even though the PAD of the upper canopy is low in winter compared to summer, there are relatively few gaps in the forest without any branches at all, effectively limiting the number of first order returns reaching the ground.

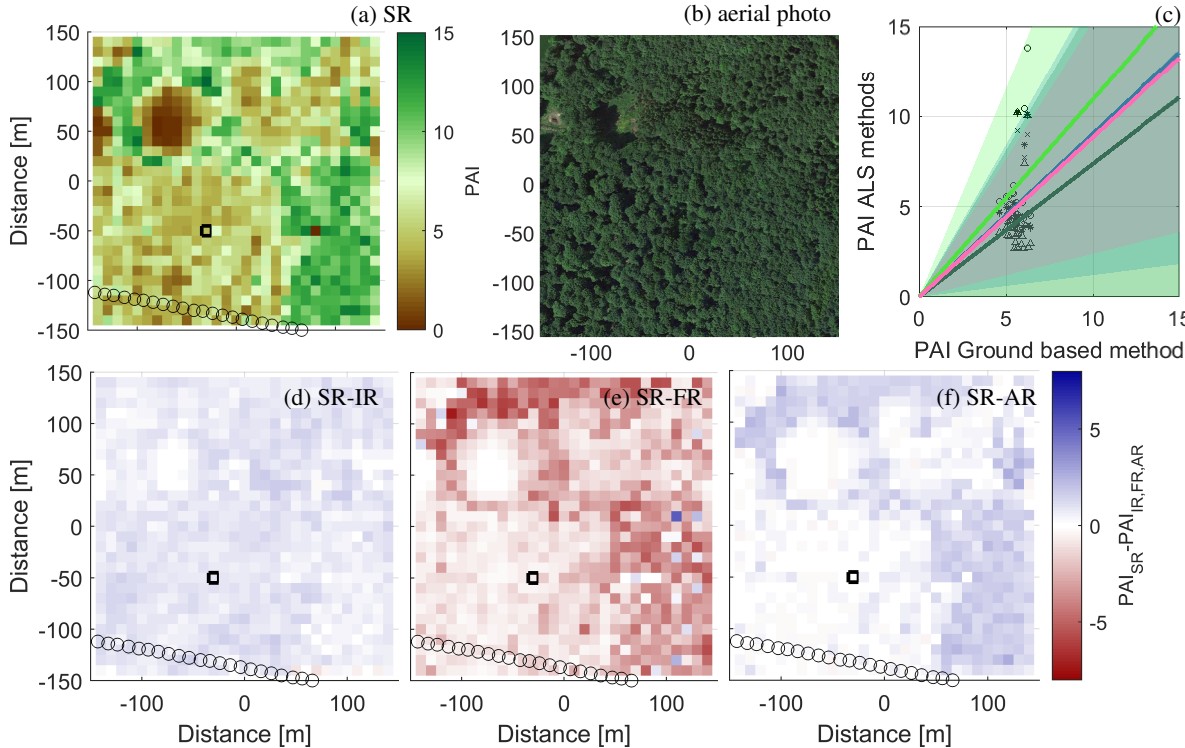

**Figure 3.** PAI estimates from the summer scan at the dense beech forest Falster site: $a$) SR method, $b$) aerial photo, $c$) scatterplot of the ground-based estimates and the ALS methods. The lines show linear regression forced through zero with shading indicating 95 % confidence level (1.96 times the standard error). SR is represented by blue and stars, IR is represented by dark green and triangles, FR is represented by bright green and circles and AR is represented by pink and crosses. The circles in the maps displays the positions of the ground based measurements. $d$)-$f$) shows the difference between SR and the reference methods IR, FR and AR respectively. The black square marks the edges for the data used in Section 4.2.

## 4.2   Single grid cell PAD and sensitivity to ground albedo

In order to illustrate how the ALS methods work, a grid cell encapsulating a single tree was selected. The location of the grid cell is highlighted in Fig. 3 and 4 with a black square. Fig. 5 shows the used reflections and the resulting PAD for the four different methods. The top row shows the summer results, whereas the bottom row shows the winter results. The size of the markers in the plots (a)-(d) and (f)-(i) indicates the weight that the algorithm gives to the reflections in the four different methods. Since intensity is not regarded in the FR and AR methods, all markers in (c)-(d) and (h)-(i) have the same size. As the SR method restricts the weight of the return, less difference in marker size are seen in plots (a) and (f), whereas more variability is seen for the IR method, where the reflections are not scaled. The first reflections are marked in blue, the second in red and the third reflection in black.

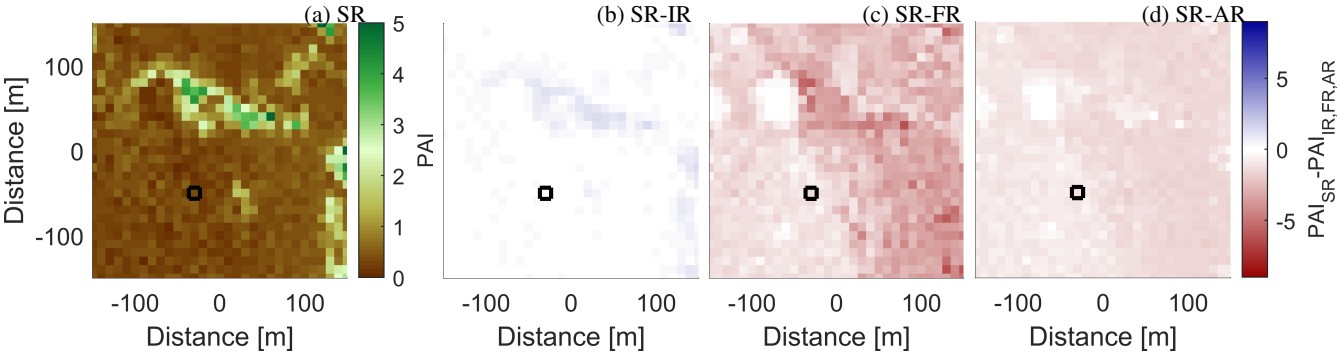

**Figure 4.** PAI estimates from Falster winter scan by various methods. *a*) show PAI in a 10 m × 10 m square grid based on the SR method. *b*), *c*) and *d*) shows the difference between IR and the reference methods IR, FR and AR respectively. The black square marks the edges for the data used in Section 4.2.

There is a marked difference between the lidar returns in the summer and winter scan. In summer (top row), the majority of the returns come from the upper part of the foliage, predominately by first and second order returns of relatively high backscatter intensity. In winter (bottom row), the intensity of the returns from the upper canopy is much lower, and the only returns with high intensity comes from the stem and larger branches, as well as the ground.

The reason for the high PAI values in FR and AR relative to SR and IR in the case of Falster winter becomes apparent from Fig. 5. The ground returns used in the AR method are often higher-order returns, which are not included in FR. As seen in Fig. 1 b, many of the gaps in the winter canopy are smaller than the  40 cm beam width for the Falster winter scan, making first order returns in the lower canopy less likely. The data in Table 2 confirms that higher order returns are more prevalent in Falster winter than the other two scans. The AR method include higher order returns, but neither FR nor AR takes into account the low intensity in the returns from the upper canopy, ultimately leading to a relatively higher estimate of PAD and PAI. Compared to the winter PAI reported in Dellwik et al. (2014) from ground-based observations, the average PAI in FR (see Table 3) was more than four times higher and for AR more than two times higher.

The way first order ground returns are treated in the methods that use the intensity of the back scatter (SR and IR) leads to a different sensitivity regarding the ground albedo. The difference in scaling technique between the SR and IR methods is visible in Fig. 5, particularly in the winter scan, through the dominance (large size) of the ground reflections in IR. Since SR uses scaling of each pulse, the intensity of a higher order ground reflection is always smaller than that of a single first reflection, no matter the actual back scattered intensity.

The sensitivity to ground albedo was investigated through artificially changing the intensity values of the ground returns according to Section 3.3. For the open pine forest of the tile at the Norunda site (Fig.2, the mean difference in sensitivity was large, with SR showing only 0.4 % change of PAI for a 10 % change in ground return intensity. The corresponding number for IR was 5.3 %. For the beech forest tile taken during the summer (Fig. 3), the sensitivity to a 10 % change in ground return intensity was 0.8 and 2.7 %, for SR and IR, respectively. For the bare trees in winter the corresponding values were 2.4 and

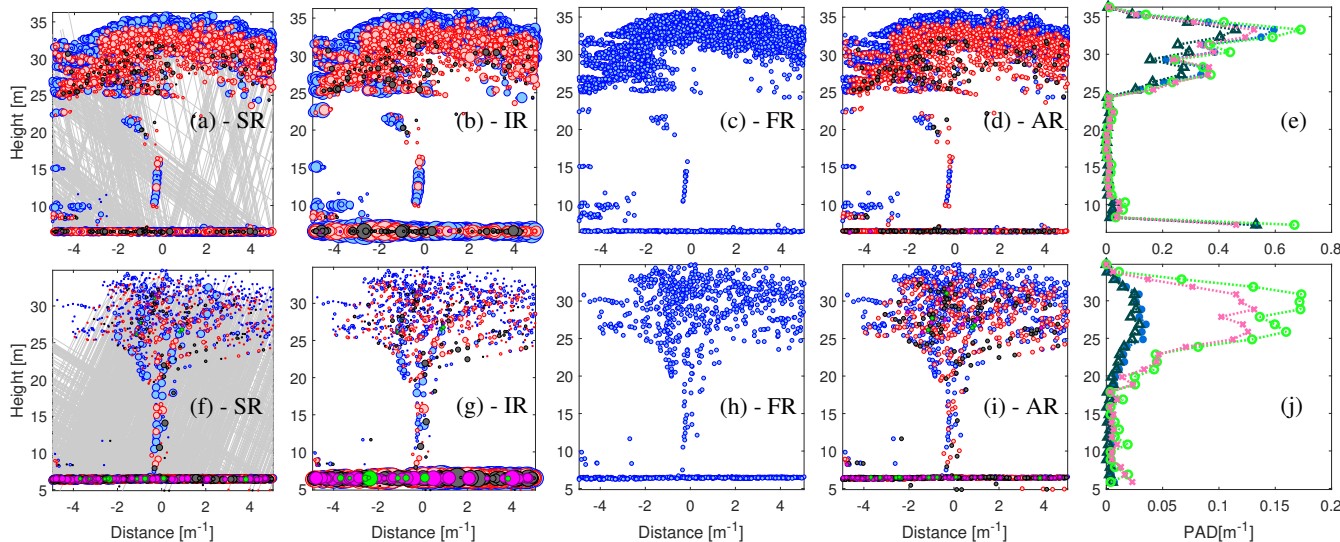

**Figure 5.** Backscatter reflections from within a 10 m × 10 m grid box, taken from the Falster summer (top row) and winter (bottom row) scans. The color indicate return number: First returns - blue, second returns - red, third returns - black, fourth returns - purple and fifth returns - green. The size of the marker indicate weight of the return according to the four methods. Scaled return number, SR ($a$ and $f$) is determined from Eq. 7, for IR ($b$ and $g$) it is scaled by the medium intensity of the first order canopy returns in the grid box and for FR and AR ($c$, $d$, $h$ and $i$), all returns are weighted the same. Note that the scale changes between the summer and winter. The grey lines indicate the path of the lidar beam between first and higher order returns. Thus, lines are only shown for beams which has more than one return. $e$ and $j$ shows the corresponding PAD for summer and winter scan respectively, SR is represented by blue stars, IR is represented by dark green triangles, FR is represented by bright green circles and AR is represented by pink crosses. The location of the grid cell can be seen in Fig. 3 and 4.

6.0 %. The larger difference in Norunda comes from the larger share of first order ground returns (see Table 2), owing to the more open structure of the forest. Compared to the absolute differences between the methods, the albedo sensitivity is relatively small, however, indicating systematic differences over the areas.

### 4.3 Sensitivity to grid size

Since the canopy elements are heterogeneously distributed on all scales that are suitable for gridding when PAD and PAI are calculated, the homogeneity assumption used when applying the Beer-Lambert law is violated. To illustrate the effect of that violation, the mapping of PAI from $\sum_{\text{ground}} R_i / \sum_{\text{all}} R_i$ by Eq. 9 is shown in Fig. 6 in the x-y plane. Jensen's inequality is seen from the strong curvature of Eq. 9, which has the effect that an even distribution in $\sum_{\text{ground}} R_i / \sum_{\text{all}} R_i$ becomes positively skewed, with the long tail on the high PAI side. Lowering the resolution means narrowing the distribution of $\sum_{\text{ground}} R_i / \sum_{\text{all}} R_i$, thereby shortening the long tail more than the short tail in the PAI distribution, effectively lowering the PAI average.

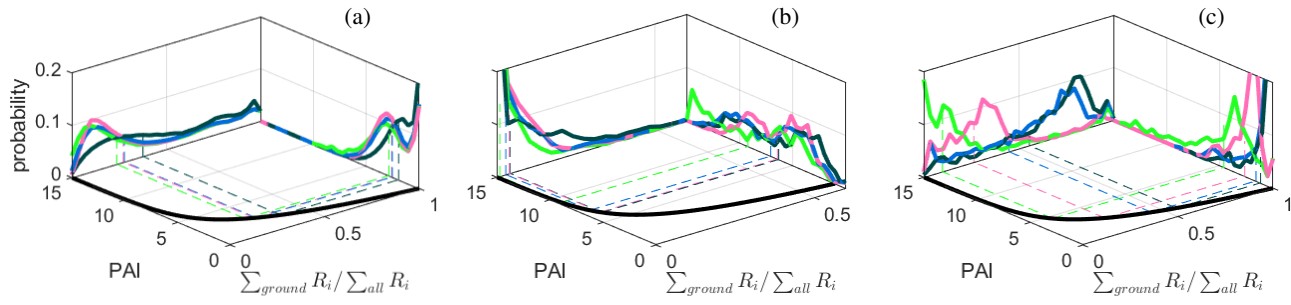

**Figure 6.** The mapping of PAI from the ratio of ground returns to total number of returns ($\sum_{\text{ground}} R_i / \sum_{\text{all}} R_i$ at $10\,\text{m} \times 10\,\text{m}$ resolution) by Eq. 9 (black line in the x-y plane). On the vertical axis are shown the respective distributions of PAI (y-z plane) and $\sum_{\text{ground}} R_i / \sum_{\text{all}} R_i$ (x-z plane), at $10\,\text{m} \times 10\,\text{m}$ resolution, for the four methods (SR -blue, IR - dark green, FR - light green and AR - pink). The dashed lines indicate the area average PAI from averaging PAI at $10\,\text{m} \times 10\,\text{m}$ resolution. For scans Norunda (a), Falster summer (b) and Falster winter (c).

**Table 3.** Values of average PAI for the two sites, using different calculation methods and grid resolutions. $\ast$: A number of PAD estimates were not possible due to fully attenuated first returns before the ground level. The data was filled with PAD values of 0.1. Less than 1 % of the estimates were affected except for Falster summer at a resolution of $10\,\text{m} \times 10\,\text{m}$, where 4 % were affected.

| Site and method | $10\,\text{m} \times 10\,\text{m}$ | $20\,\text{m} \times 20\,\text{m}$ | $50\,\text{m} \times 50\,\text{m}$ | $100\,\text{m} \times 100\,\text{m}$ |
|---|---|---|---|---|
| Norunda SR | 2.56 | 2.42 (0.95) | 2.35 (0.92) | 2.29 (0.90) |
| Norunda IR | 1.95 | 1.80 (0.92) | 1.70 (0.87) | 1.64 (0.84) |
| Norunda FR | 2.87$^\ast$ | 2.69 (0.94) | 2.58 (0.90) | 2.51 (0.87) |
| Norunda AR | 2.60 | 2.51 (0.96) | 2.45 (0.94) | 2.41 (0.93) |
| Falster summer SR | 7.09 | 6.58 (0.93) | 5.94 (0.84) | 5.68 (0.80) |
| Falster summer IR | 6.34 | 5.74 (0.90) | 5.03 (0.79) | 4.75 (0.75) |
| Falster summer FR | 9.23$^\ast$ | 8.68$^\ast$ (0.94) | 7.39 (0.80) | 6.77 (0.73) |
| Falster summer AR | 6.45 | 6.07 (0.94) | 5.62 (0.87) | 5.46 (0.85) |
| Falster winter SR | 1.60 | 1.45 (0.91) | 1.31 (0.82) | 1.23 (0.77) |
| Falster winter IR | 1.17 | 1.01 (0.86) | 0.86 (0.74) | 0.79 (0.68) |
| Falster winter FR | 4.63$^\ast$ | 4.13 (0.89) | 3.76 (0.81) | 3.48 (0.75) |
| Falster winter AR | 2.66 | 2.59 (0.97) | 2.50 (0.94) | 2.48 (0.93) |

The sensitivity of the routines to grid size was investigated by evaluating PAD and PAI for the three data sets with different horizontal resolution, ranging from $10\,\text{m} \times 10\,\text{m}$ to $100\,\text{m} \times 100\,\text{m}$. Table 3 summarizes the sensitivity to grid size, showing that for a magnitude change in grid size, the change in PAI ranges from between a few percent to more than 30 % depending on site and method. The AR method was found to be the least sensitive to grid size changes, followed by the SR method. The FR method was the most sensitive. The results indicate that IR and FR sees more heterogeneity than the AR and SR methods. This is also observed in Fig. 6, where the distributions of $\sum_{\text{ground}} R_i / \sum_{\text{all}} R_i$ for IR is skewed towards high values (low PAI) and

FR towards low values (high PAI). The observation in Section 4.1 that the AR method predicts less variations in PAI spatially, mainly by underprediction in dense areas relative to the other routines can also be seen in Fig. 6, particularly in Fig. 6 (c), Falster winter, where the distribution of $\sum_{\text{ground}} R_i / \sum_{\text{all}} R_i$ is much less skewed for AR then for the other methods. As also noted in Section 4.1, the SR method behaves as the FR method in sparse regions and like the IR method in dense regions,

5    thus avoiding overprediction in the most dense areas and underprediction in the least dense areas. As shown in Section 4.2 the weight of the first order ground returns is limited in the SR method, while for IR the reflectively of the ground leads to lower PAI estimates, thereby widening the overall estimated PAI distribution.

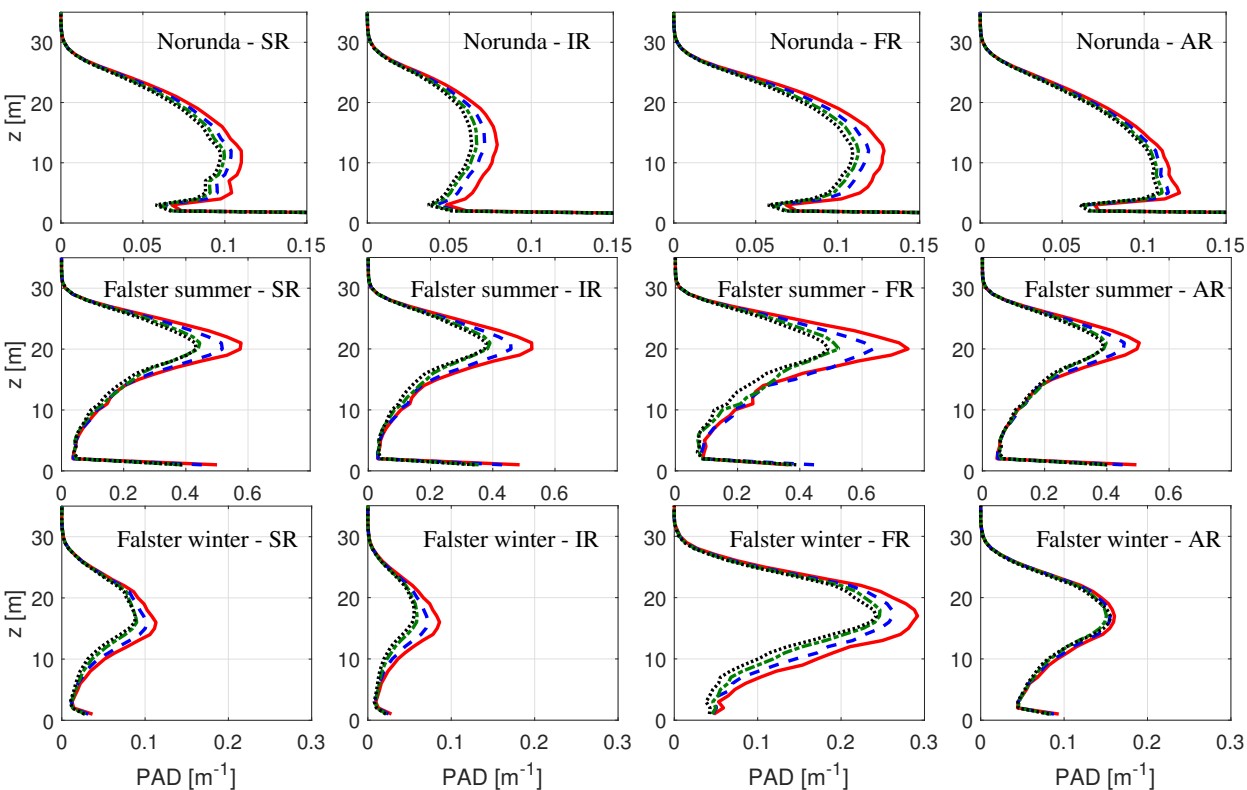

**Figure 7.** Average PAD profiles for the Norunda (upper), Falster summer (middle) and Falster winter (lower) sites. The averages are based on $10\,\text{m} \times 10\,\text{m}$ (red full line), $20\,\text{m} \times 20\,\text{m}$ (blue dashed line), $50\,\text{m} \times 50\,\text{m}$ (green dash dotted line) and $100\,\text{m} \times 100\,\text{m}$ (black dotted line) resolution. From methods SR, IR, FR and AR (left to right).

Average PAD profiles for the three data sets under varying grid size is seen in Fig. 7, where the data in Table 3, when split up into vertical profiles, reveal some differences between the methods. The difference in PAD magnitude follow that of PAI,

10   with magnitudes particularly contrasting for Falster winter, but there are also some differences in the shape of the profile. In Norunda and Falster summer, the SR and AR methods appear very similar from Table 1 and Fig. 2 (a), (d) and (g) as well as Falster summer (g). However, studying the PAD profiles, the SR method shows a less bottom heavy profile, with a PAD maximum above 10 m height, in the canopy of most trees, whereas the AR method has the maximum PAD at around 5 m. For

Falster summer, the difference between SR and AR is mainly in the dense regions (see, Fig. 3 (a) and (d)) which in the average profiles translates to a less pronounced upper canopy peak in AR. In the Falster winter case, the profiles reveal a difference between the intensity based SR and IR, with maximums at lower height and the FR and AR with maximums at higher heights. This is consistent with the case study in Section 4.2, where a lack of intensity information regards the upper canopy reflections with higher weight.

## 5  Discussion

### 5.1  Comparison between ALS routines

In general, the methods resulted in similar spatial patterns and magnitudes of PAD. Systematic differences were however found for specific conditions. For winter (leafless conditions) the results showed differences between area averaged PAI of more than 400 %. Both FR and AR showed over predictions relative to estimates in Dellwik et al. (2014), and furthermore, values by FR and AR exceeded the maximum for beech trees reported in Bréda (2003). In addition, the ratio of winter to summer PAI was in excess of 0.4 for both FR and AR, also outside the reported range in Bréda (2003). We attributed this to the high weights of reflections from the upper canopy, a result that follow from the lack of intensity information in connection to the tiny frontal area of branches to the footprint area of the lidar. The FR method further showed problems in the summer scan of the same forest, with low amount of ground returns making PAI estimates very high and sometimes impossible due to lack of ground returns (argument in the logarithm of Eq. 3 becomes zero). The result indicates that using the intensity information provides a key benefit when the lidar footprint is large compared to common gap sizes in the forest. On the other hand, the results also show sensitivity to ground albedo when the intensity information is used. This sensitivity was present in both IR and SR method, but the effect in SR was limited to between 7.5 and 40 % of the sensitivity in IR. Overall, the scaling technique introduced in SR limits effect related to ground albedo and foot print to gap size ratio, which indicates it might be less sensitive to stand characteristics as well as scanning parameters such as flight height and scan density. The present study included needle and broadleaf trees for in-leaf and leafless conditions, and dense helicopter scan with small lidar footprint as well as sparse scans with a relatively larger footprint, but future studies systematically varying scan characteristics in different types of forests could help further quantify differences between methods.

The method of Vincent et al. (2017) is similar to the method presented herein, and thus potentially carries many of the advantages of the SR method. However, their use of weights based on the *average* intensity per return number and number of returns, is a key difference from using the actual intensity fraction of the return within the pulse. In their study, Vincent et al. (2017) found that return number within the pulse and total number of returns in the pulse described 51 % of the variance in intensity, which indicates that using average intensity numbers omits roughly half of the variations that is captured by using pulse-specific intensity information. For the data sets presented in this study, the average intensity roughly splits evenly among the returns in the two full-leaf scans, but the leafless scan was very different, for which the average ground return consistently was more than twice as intense as any of the canopy returns, regardless of the number of returns. When we tested the difference between the SR method using individual and average weights the absolute relative difference in PAI estimates between the two

was less than 5 % in the full leaf scans, but for the winter scan the difference was on average 25 %. Hence, if possible, the use of individual weights should be preferred, especially during the leafless state.

In cases of strong heterogeneity and focus on actual PAD, as opposed to effective PAD, the method presented in Detto et al. (2015) provides an alternative to the approximation Beer-Lambert law, and future research in that direction is interesting, especially with increasing access to computational power.

## 5.2 Effects of resolution

How to optimally select resolution for PAI/PAD estimation by ALS is still an open question. From Table 3, it is clear that a coarser resolution leads to a systematically lower PAI. The results agree with Boudreault et al. (2015), who presented a study for grid resolution using the FR algorithm in a spruce forest, varying resolution from 5 m × 5 m to 30 m × 30 m. The results also agree with the analysis of Dufrêne and Bréda (1995), who presented a comparison between LAI from litter collection and observations from ground-based PAI observations using the Plant Canopy Analyzer (LI2000, LiCor Inc., Nevada), which is based on light penetration from five concentric rings. The PAI readings from the Plant Canopy analyzer systematically decreased in magnitude when rings at higher zenith angles were included in the estimate. The inclusion of more rings at high zenith angles can directly be translated to an enlargement of the measurement footprint, which makes the results by Dufrêne and Bréda (1995) comparable to the ALS based results on gridding resolution from this study. As stated above, Jensen's inequality for convex functions explains these systematic changes. Hence, when estimating PAI for heterogeneous forests from instruments that are based on the exponential decay of radiation into the canopy, the results can be expected to be resolution dependent. It should also be noted that results shown in Fig. 6 demonstrate that grid size sensitivity is dependent on the magnitude of PAD/PAI, with dense forests having a more non-linear mapping from ALS to PAD/PAI than less dense, and that this dependence enhances grid size sensitivity. Additionally, as discussed above, forest density in combination with the scan density (number of returns per $m^2$) sets an upper limit to the possible resolution. Whereas the results in Table 3 shows that systematic changes with resolution occurred for all four tested methods, there was significant variability between the tested methods, where FR always showed the highest sensitivity and AR and SR the lower showed the lowest sensitivity.

The optimal resolution likely depends on the application. In wind modelling there is for example the optimal ratio of resolution of PAD field and flow field to consider, where the resolution of the flow field puts an upper limit to the resolution to the PAD field. Ivanell et al. (2018) concluded that for a predominantly forested area, where the drag force by the forest on the wind was modelled by means of PAD profiles, the value of the surface roughness below the canopy had very little impact on the flow above the forest. This implies that for wind modelling, uncertainty in the lower part of the canopy is acceptable. Further research in how to optimally choose grid resolution is needed.

## 6 Conclusion

We presented a new method to calculate PAI and PAD from airborne laser scans (ALS), which uses all available reflection attributes commonly stored in the ALS datasets, and combines benefits from earlier published algorithms. In order to evaluate

the performance of the new method, it was used together with three reference methods on sites with contrasting characteristics. The results of the new routine showed no marked weakness with respect to sensitivity to forest characteristics, grid resolution, beam attenuation and ratio of ground to canopy returns, whereas all reference methods were challenged in either of those categories. In summary, the results indicate that the new algorithm is robust and would therefore perform well for a wider range of vegetation types than tested here.

The new routine is applicable provided that each data point in the data set contains information on the number of returns in the pulse, the return number of the specific point and that the data is ordered such that consecutive reflections from the same pulse are grouped in the data set. For the ALS datasets that we have used in this study, this criterion has been fulfilled.

The results showed a dependence on the grid resolution of the derived PAI and PAD fields that was systematic for all methods. The grid sensitivity was linked to the violation of the homogeneity assumption in the Beer-Lambert law, which can also be explained by the Jensen's inequality for convex functions. The grid resolution sensitivity is varying between the methods (from 7 and 32 % reduction in PAI for a magnitude change in grid size), where the new method showed the second least sensitivity.

Despite the identified issues with airborne lidars for the determination of plant area and plant density, the similarities between the four tested methods highlight the prospect of obtaining PAD and PAI with great spatial coverage, that are both precise and accurate, especially in regards to the uncertainty levels in established ground based methods (Yan et al., 2019). The use of airborne lidar scans to derive maps of PAD and PAI also provides a promising way to greatly reduce the subjectivity, and likely uncertainty, of any atmospheric modelling that involves air-canopy interactions, such as drag, heat transfer and gas exchange.

*Code availability.* TEXT

A Python (Python Software Foundation, https://www.python.org/) as well as MATLAB (Mathworks, Inc.) implementation of the algorithm that outputs PAD, PAI, forest height and ground elevation from laserdata in the .las format has been developed and is freely available at https://github.com/johanarnqvist/ALS2PAD.git/.

*Author contributions.* All authors contributed to study design, writing the manuscript, preparing data sets, analysis and discussion of the results. J. Arnqvist developed the new routine and had the main responsibility for writing, data analysis and figure preparation.

*Competing interests.* None of the authors have competing interests.

*Acknowledgements.* We acknowledge Irene Lehner for providing ground based measurements at the ICOS Sweden station Norunda. Dr Ferhat Bingöl is acknowledged for the photos in Fig. 1 and Poul Therkild Soerensen is acknowledged for the reference measurements of PAI at the Falster site. Part of this work was conducted within STandUP for Wind, a part of the STandUP for Energy strategic research

framework (Sweden). The Independent Research Fund Denmark, via the The Single Tree Experiment project [Grant No. 6111-00121B], and the European Commission, via the New European Wind Atlas Project, are also acknowledged for funding.

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
