# Peer review of "Robust processing of airborne laser scans to plant area density profiles"

_Biogeosciences, 2020_

## Referee Comment (RC1) · Anonymous Referee #1 · 19 May 2020

The manuscript, entitled "Robust processing of airborne laser scans to plant area density profiles", presents a new approach to estimate Plant Area Density (PAD) using airborne lidar data. This type of research is highly needed and a topic of interests in Biogeosciences, as we need to improve our understanding of plant area allocation, vertical profiling, and its relationship to Leaf Area Density (LAD) that is related to many biochemical functions of trees. We also need to overcome some data and model challenges by using airborne lidar for PAD estimation. However, the current version is insufficient for publication with two major concerns as listed below:

1. No accuracy assessments/ statistical analysis to validate the improvement of the new proposed results. The new proposed approach applied scaled intensity information to Beer-Lambert law to improve the PAD estimation, and compared it to three other

[Figure]

methods (IR: un-scaled intensity return method, FR: first return method, and AR: all return method). And it also has LAI2000 data as field measurements. However, there is no statistical analysis / accuracy assessment, such as r-squared and paired t-test to show the new results preforms better than the other three. The only discussion on this was based on the visualizations on Fig 2 on Page 8. The results of the proposed method is very similar to the results of AR on Fig 2. So the conclusion of this study is arguable without statistical tests for comparison and statistical validation with field measurements.

2. The writing needs improve. Equations should go to the method section. And some equations are redundant e.g. Eq. 4, 5, and 6 are basically the same equation with different inputs. The description of new approach is vague and confusing. And there is no explanation why the scaled intensity is able to fix the issue of ground PAD reflections in IR. And also, lots of the statements have grammar issues and debatable context, which made the manuscript hard to read.

―――――――――――――――――

---

## Author Comment (AC1) · 1 Jun 2020

We thank Referee #1 for a constructive review. While we prefer to wait for more Referee comments before adjusting the manuscript and properly addressing all issues raised, we here take the opportunity to give an initial response on the criticism raised mainly regarding the comparison with ground-based observations.

"No accuracy assessments/ statistical analysis to validate the improvement of the new proposed results."

Our paper is based on the small discovery that it is possible to trace nearly all (more than 99%) canopy reflections to the emitted laser pulse from which they originate. To illustrate this aspect, we have added the grey lines in Fig. 5. To be able to trace the

pulse throughout the canopy could lead to future improvements concerning comparison with ground-based techniques, since it provides more knowledge on how the pulse is affected by the complex canopy geometry. Hence, the new method more closely resembles the very early works for estimating leaf area by sampling canopy elements using long poles (Wilson, 1959), than other airborne lidar methods. Rather than poles, we here use the geometric lines defined by the lidar pulse. Based on this discovery, our main scope in the study is to introduce a new method for a physically sound plant-area-density estimation and evaluate the resulting magnitudes to other airborne lidar methods. To our knowledge, such a comparison using several ALS-> PAD methods is the first of its kind. A thorough comparison with ground-based observations would need to address the well-known weaknesses in current ground-based techniques (in example, the unknown footprint of the PCA-2000 by LiCor as well as the assumption of homogeneity) and the issue of canopy elements shading each other using fish-eye lense photography (Yan 2019), and this is beyond the scope of the current paper. In our study, we have instead chosen to focus on the demonstration of resolution dependence and add insight using the well-known framework of Jensen's inequality. It would be easy to add a more careful comparison with the ground-based reference method, such as a paired t-test or an r2 estimate, but given the referee's second comment concerning readability, we suggest to rather tone down this comparison and move it to an Appendix. The positions where we have access to reference data (seen in Fig. 2 and 3) all lie within relatively low PAI regions, and there is not much variation in PAI magnitude. This is particularly evident in Fig. 3. As Referee #1 pointed out, there does not seem to be much difference between SR and AR methods based on comparison with ground-based methods, and this is true. However, by studying Fig. 2 and Fig. 3 it is evident that there are areas with very large differences between SR and AR. The reason for this is highlighted in Fig. 5 and accompanying discussion, and we will work to clarify this point in the next version of the paper. In summary, we share the referee's view that a quantitative comparison would be desirable, but argue that given current uncertainties in the referred ground-based techniques, such a detailed comparison adds

too little value to represent a main part of the paper.

"And there is no explanation why the scaled intensity is able to fix the issue of ground PAD reflections in IR"

Hopefully the revision will clarify our explanation, but in the meantime we would like to direct attention to Fig. 5 a, b, f and g, discussion along that figure and in Sec. 2.3, line 15. Since the IR values are scaled for each individual pulse the value of any individual pulse has an upper limit of 1. For the case when all the pulses have returns that come from both canopy and ground, the improvement of SR relative to IR will be minor, but in case of heterogeneities, there will be pulses that only have ground reflections despite that the there is a canopy above since the beam is tilted. . In this case, there will be a larger difference between the two methods. The value of first-order ground returns will always be 1 in SR, but in IR the value will be that of the backscattered intensity (compare blue points at the ground in Fig. 5). As pointed out previously (Wagner et al. 2008), this can lead to a bias in the PAI/PAD estimates if the albedo of vegetation and ground is different. In summary, if there is a considerable amount of 1st order ground returns, and the ground has a different albedo to the laser beam, the SR method should limit that bias and be superior to the IR method.

Wilson, J.W. Analysis of the spatial distribution of foliage by two-dimensional point-quadrats. New Phytol. 1959, 58, 92–99.

Yan, G., Hu, R., Luo, J., Weiss, M., Jiang, H., Mu, X., Xie, D., and Zhang, W.: Review of indirect optical measurements of leaf area index: Recent advances, challenges, and perspectives, Agricultural and Forest Meteorology, 265, 390 – 411, https://doi.org/https://doi.org/10.1016/j.agrformet.2018.11.033, http://www.sciencedirect.com/science/article/pii/S0168192318303873, 2019.

---

## Referee Comment (RC2) · Anonymous Referee #2 · 25 Jun 2020

TArnqvist, Freier, and Dellwik introduce a new 'scaled ratio' method for estimating plant area density from airborne scans and compare it to established methods in two temperate forests. The algorithm performs similarly to established algorithms with some notable improvements. The tone of the critique of existing methods was unacceptable and needs to be changed. Methods are designed to be improved upon, not disparaged. The fundamental challenge with the comparison is that there is no basis for determining if one method is better than another, which can really only be determined using computer simulations unless 'true' PAI can somehow be calculated in a field setting. This doesn't diminish the publishablility of the results, but further emphasizes that the tone of the comparison needs to change and that the benefits of the present approach need to be mediated somewhat.

P1 L11: 'never' is a bold statement. There are a number of passages where the skill of the new model is oversold, which will diminish the impact of the present analysis and leave it less clear where further model improvements should be focused.

P1 L21: really plant area density is being measured here, which is fine, just please be clear about it at the start.

Please provide citations for the statement on page 2 Line 7-8.

P3 L 27 do the authors mean Eq. 4-6?

P4 top line yes, but with enough beams something will go through. Is there a good reference for the notion that first returns is 'problematic'? (avoid this word, and 'problem' if possible, also later in the paragraph; this paragraph is poorly references). The title of section 2.3 is entirely too harsh. These researchers worked hard on these methods and the title diminishes their efforts. "combine the benefits" on p. 4 L. 30 is much better. No method is perfect.

P. 6 L. 5: atmospheric scattering will apply to downward and upward-traveling beams.

P6 L19 typo on 'therefore'

Figure 2 legend: the bright green text (and figure symbols) is not easy on the eyes.

Figure 2 and 3: histograms would also be interesting to compare.

P. 10 L. 22: without a 'true' value it isn't accurate to say that one method or another is an 'overestimation'. This could however be determined in a simulation.

p. 15: remove 'dubious'.

―――――――――――――――――――

---

## Author Comment (AC2) · 15 Jul 2020

**Answers to referee 1**

Arnqvist et al.

**Correspondence:** Johan Arnqvist (johan.arnqvist@geo.uu.se)

We appreciate the time the referee has taken into reading the manuscript. We agree with most of the reviewer comments and beleive that the text have been improved in many places. In the following we address the comments made and point out changes in the revised manuscript. Please also read the comments to Referee # 2 for other revisions of the manuscript.

**Comments**

5      1. *No accuracy assessments/ statistical analysis to validate the improvement of the new proposed results. The new proposed approach applied scaled intensity information to Beer-Lambert law to improve the PAD estimation, and compared it to three other methods (IR: un-scaled intensity return method, FR: first return method, and AR: all return method). And it also has LAI2000 data as field measurements. However, there is no statistical analysis / accuracy assessment, such as r-squared and paired t-test to show the new results preforms better than the other*

10      *three. The only discussion on this was based on the visualizations on Fig 2 on Page 8.*

We acknowledge the lack of statistical measures in the original manuscript and have updated the revised manuscript with more quantification. As we argued in the initial response, our goal is not a validation with with ground-based observations (which would also need to quantify weaknesses in current ground-based techniques). The aim with our study is instead to present the new method and illustrate how it combines benefits of previously published ALS to PAD methods. The improve-

15      ments are in short; limiting the influence of the albedo of first order ground returns (relative to the IR method), limiting attenuation problems in dense areas (relative to the FR method) and making estimates more in line with published values of leafless canopies (relative to FR and AR methods). In the revised manuscript we have worked to further emphasize and quantify that. Our changes include:

– Addition of 95% confidence bounds to the regression lines.

20      – Added a discussion based on winter PAI values for Beech forests from the litterature to assess the PAI magnitudes of the different methods for the winter scan.

– Replaced the map plots for the reference methods to difference plots between the new method and the reference methods (see Fig. 1, 2 and 3 below)

– Clarified the purpose of Fig 5 and accompanying discussion.

- Added a sensitivity study to the intensity values of ground returns to quantify that IR is more sensitive than SR (2.5-13 times as sensitive). Please see answer #2 for more information.

[Figure]

**Figure 1.** PAI estimates from Norunda by various methods. *a*) show PAI in a 10 m × 10 m square grid based on the SR method. *b*) shows an aerial photo over the site. *c*) is a scatterplot of the hemispheric photo estimates and the ALS methods. The lines show linear regression forced through zero with shading indicating 95 % confidence level (1.96 times the standard error). SR is represented by blue and stars, IR is represented by dark green and triangles, FR is represented by bright green and circles and AR is represented by pink and crosses. The circles in the maps displays the positions of the ground based measurements. *d*)-*f*) shows the difference between IR and the reference methods IR, FR and AR respectively.

> 2 *The results of the proposed method is very similar to the results of AR on Fig 2. So the conclusion of this study is arguable without statistical tests for comparison and statistical validation with field measurements.*

5    As noted in the initial response, we would like to direct interest to the single grid cell illustration, for clarification of the differences. In order to further highlight the contrasts between the different methods we have replaced Fig 2-4, b-d with difference plots. The main difference is in the winter scan of the deciduous forest. As can be seen in Fig. 6 and Table 2, both the median and the mean value of the PAI in the winter scan is high for FR and AR relative to SR and IR. In comparison to published values of winter PAI for beech forests, both FR and AR are above the maximum finding in Bréda (2003). In addition

[Figure]

**Figure 2.** PAI estimates from Falster summer by various methods. $a$) show PAI in a 10 m × 10 m square grid based on the SR method. $b$) shows an aerial photo over the site. $c$) is a scatterplot of the hemispheric photo estimates and the ALS methods. The lines show linear regression forced through zero with shading indicating 95 % confidence level (1.96 times the standard error). SR is represented by blue and stars, IR is represented by dark green and triangles, FR is represented by bright green and circles and AR is represented by pink and crosses. The circles in the maps displays the positions of the ground based measurements. $d$)-$f$) shows the difference between IR and the reference methods IR, FR and AR respectively. The black square marks the edges for the data used in Section 4.2.

to that, the ratio of winter to summer PAI is in excess of 0.4 for both FR and AR, also that above the maximum presented in Bréda (2003). We have clarified the relative differences in the results section and added into the discussion:

"The winter estimates of average PAI in FR and AR exceed the maximum value for Beech trees reported in Bréda (2003). In addition to that, the ratio of winter to summer PAI is in excess of 0.4 for both FR and AR, also that above the maximum presented in Bréda (2003)."

> 3 *The writing needs improve. Equations should go to the method section. And some equations are redundant e.g.*
> *Eq. 4, 5, and 6 are basically the same equation with different inputs.*

We have created a new subsection, *3.3 Evaluation parameters*, where the comparison methods of *Spatial differences, Comparison with ground based methods, Sensitivity to ground albedo* and *Sensitivity to grid size* are explained. This facilitated a

[Figure]

**Figure 3.** PAI estimates from Falster winter scan by various methods. *a*) show PAI in a 10 m × 10 m square grid based on the SR method. *b*), *c*) and *d*) shows the difference between IR and the reference methods IR, FR and AR respectively. The black square marks the edges for the data used in Section 4.2.

move of method oriented paragraphs from the results section to the method section and, we hope, clarified the structure of the manuscript.

Eq. 4-6 serves to explain the foundation of each method, and is therefore important for the remainder of the text. Even though the differences may seem small, in fact they are important, as can bee seen in the new figures (also included here as Fig. 1, 2 and 3). Furthermore, the style is in line with previous literature (Hopkinson and Chasmer, 2009), and we have therefore kept them as they are.

4 *The description of new approach is vague and confusing. And there is no explanation why the scaled intensity is able to fix the issue of ground PAD reflections in IR.*

We have made revisions to the text clarifying the SR method further. To quantify the sensitivity to difference in ground albedo to canopy albedo we have included results from artificially varying the ground return intensities. This enables us to say that the sensitivity of SR is significantly less than for IR. We include the updated text below.

In the methods section:

"In order to investigate the sensitivity to ground albedo, a test was constructed where the intensity values of returns classified as ground were manipulated. Intensity from ground returns were increased or decreased by a factor of 1.1 or 0.9 after which PAI was calculated and compared to PAI derived from the original data set. The manipulation only affects SR and IR, as FR and AR excludes intensity information. Both SR and IR showed only minor differences in sensitivity between a 10 % decrease and a 10 % increase in ground intensities, so results are presented as sensitivity to a 10 % ground intensity alteration.

And in the results section:

"The way first order ground returns are treated leads to a difference in ground albedo sensitivity between SR and IR. The difference in scaling technique between the SR and IR methods is visible in Fig. 5, particularly in the winter scan, through the dominance of the ground reflections in IR. Since SR uses scaling of each pulse, the intensity of a higher order ground reflection is always smaller than that of a single first reflection, no matter the actual back scattered intensity. This can be an advantage,

as some surfaces may very effectively block the pulse despite sending relatively low back scattered intensity. Furthermore, the first order ground returns in SR all have the same weight, whereas in IR the weight is directly proportional to their share of the total backscattered intensity.

The sensitivity to ground albedo was investigated through artificially changing the intensity values of the ground returns according to Sec. 3.3. For the open pine forest in Norunda, the difference in sensitivity was large, with SR showing only 0.4 % change of PAI for a 10 % change in ground return intensity. The corresponding number for IR was 5.3 %. For the summer Beech forest in Falster, the sensitivity to a 10 % change in ground return intensity was 0.8 and 2.7 % respectively for SR and IR. For the bare trees in winter the corresponding values were 2.4 and 6.0 %. The larger difference in Norunda comes from the larger share of first order ground returns, owing to the less dense forest."

*5 And also, lots of the statements have grammar issues and debatable context, which made the manuscript hard to read.*

We have revised the manuscript to clarify our statements and remove incorrect grammar.

The comments from the reviewer have certainly helped us to improve our manuscript and we hope that the comments have been taken into consideration satisfactorily.

**References**

Bréda, N. J. J.: Ground based measurements of leaf area index: a review of methods, instruments and current controversies, Journal of Experimental Botany, 54, 2403–2417, https://doi.org/10.1093/jxb/erg263, https://dx.doi.org/10.1093/jxb/erg263, 2003.

Hopkinson, C. and Chasmer, L.: Remote Sensing of Environment Testing LiDAR models of fractional cover across multiple forest ecozones, Remote Sensing of Environment, 113, 275–288, https://doi.org/10.1016/j.rse.2008.09.012, http://dx.doi.org/10.1016/j.rse.2008.09.012, 2009.

---

## Author Comment (AC3) · 15 Jul 2020

**Answers to referee 2**

Arnqvist et al.

**Correspondence:** Johan Arnqvist (johan.arnqvist@geo.uu.se)

We appreciate the time the referee has taken into providing constructive comments, which we have taken into consideration revising the manuscript. In the following we address the comments made and point out changes in the revised manuscript. Please also read the comments to Referee # 1 for other revisions of the manuscript.

**General comment**

5 1. *Arnqvist, Freier, and Dellwik introduce a new 'scaled ratio' method for estimating plant area density from airborne scans and compare it to established methods in two temperate forests. The algorithm performs similarly to established algorithms with some notable improvements. The tone of the critique of existing methods was unacceptable and needs to be changed. Methods are designed to be improved upon, not disparaged. The fundamental challenge with the comparison is that there is no basis for determining if one method is better than another,*
10 *which can really only be determined using computer simulations unless 'true' PAI can somehow be calculated in a field setting. This doesn't diminish the publishablility of the results, but further emphasizes that the tone of the comparison needs to change and that the benefits of the present approach need to be mediated somewhat.*

First and foremost, we regret to hear that the manuscript came across as negative in terms of how the reference methods were portrayed. This was not our intention and we have made major revisions to the manuscript to remove any doubt that we
15 find the reference methods significant scientific contributions. In fact, both the IR and the FR methods were pioneering works in PAD estimation from ALS, and as such certainly represent larger contributions than the method presented in this study. Our intention was to clearly single out the specific contribution of the work presented in the manuscript, and it was with that aim in mind we made our original formulations.

We agree with the referee that a fundamental challenge is the lack of reliable field data of PAD, covering a wide range of
20 conditions, that can serve as validation data. In spite of that, we do think that there are still some theoretical considerations in which the new method may be considered an objective improvement, namely limiting the influence of the albedo of first order ground returns (relative to the IR method) and limiting attenuation problems in dense areas (relative to the FR method). We agree with the referee that it is challenging to assess whether either of the methods provide more accurate PAD magnitudes, and with that in mind we have adjusted the manuscript according to the referee's suggestions. Regarding the possibility of
25 using computer models to objectively judge the validity of the methods, this is an interesting idea, but we also see some challenges. For example, accurately constructing the computer model to represent the distribution of scattering elements, the

angle distribution of scattering elements, the albedo of different surfaces and accurately modelling thresholds for discrete return classification of backscatter.

**Specific comments**

*1. P1 L11: 'never' is a bold statement. There are a number of passages where the skill of the new model is oversold, which will diminish the impact of the present analysis and leave it less clear where further model improvements should be focused.*

We acknowledge that never is an unsuitable formulation. It was referring to the investigated data sets, not intended as a "global" never. We have rephrased the sentence.

*2. P1 L21: really plant area density is being measured here, which is fine, just please be clear about it at the start.*

We can see how the sentence may come across as confusing. While in some modelling applications PAI is the important measure (such modelling of wind drag), for others LAI is instead what is important. We do not want to mislead a reader to think that LAI is what is being measured (or rather calculated), so we have updated the sentence. There is a widespread misuse of "LAI" in the wind modelling community, when really "PAI" is what is used which also contribute to the confusion. We fully agree with the referee that the two are not the same (emphasized by the difference in summer and winter scan) and should not be confused with each other.

*3. Please provide citations for the statement on page 2 Line 7-8.*

We have added the references Solberg et al. (2006); Morsdorf et al. (2006); Richardson et al. (2009); Boudreault et al. (2015); Almeida et al. (2019); Hopkinson and Chasmer (2009) to the statement.

*4. P3 L 27 do the authors mean Eq. 4-6?*

Yes, thank you for pointing this out.

*5. P4 top line yes, but with enough beams something will go through. Is there a good reference for the notion that first returns is 'problematic'? (avoid this word, and 'problem' if possible, also later in the paragraph; this paragraph is poorly references).*

The formulation is based on the authors' experience working with nationwide and regional scan data bases in the Scandinavian countries, Germany and Spain. Those scans typically have a density of 0.5-10 points per square meter, which leads to attenuation before the ground in grid points with dense forests if the resolution is in the order of 10x10 m. In fact, the summer scan of Falster is from a separate helicopter scan and has on average 32 points per square meter, and still reach attenuation before the ground in 4 % of the grid points for the FR method at 10x10 m. This is reported in table 2 of the manuscript. Hence,

even with a large number of beams, there will be attenuation of the first order returns in dense canopies, especially if flight height is large so that the width of the beam is relatively large (for nationwide scans a typical value is around 50 cm diameter). To make the motivation clearer we have elaborated on the issue, included the reference Freier (2017) and avoided *problematic* according to the referee suggestion.

6. *The title of section 2.3 is entirely too harsh. These researchers worked hard on these methods and the title diminishes their efforts. "combine the benefits" on p. 4 L. 30 is much better. No method is perfect.*

We regret the clumsy wording. We suggest the new section heading *Potential to combine benefits of existing methods*.

7. *P. 6 L. 5: atmospheric scattering will apply to downward and upward-traveling beams.*

Regarding one-way or two-way extinction: The airplane flies over a canopy and emits a beam at angle alpha. Due to atmospheric scattering, the cross section of the beam grows with distance from the canopy and could have a size of a relatively large disk (for national scans 40 cm, Freier (2017)) when the pulse reaches the top of trees. The beam scatters in all directions and only a small part is directly reflected back to the airplane. Because the lidar beam travels at the speed of light, and the speed of the airplane is many orders of magnitude lower, the angle of the emitted beam can be considered to be equal to the angle of the reflected beam (the airplane can be considered to be frozen in space for the time it takes the lidar beam to reflect). When entering the canopy, a first part of the beam is reflected on a top canopy element, effectively cutting away a slice from the disk, which continues further into the canopy. When reaching the next canopy element, the reflected radiation passes the first top canopy element again, but to a first approximation no further radiation is removed, since this part of the disk has been removed in the first reflection. This is why almost all ALS to PAD/PAI studies do not consider two-way extinction.

Atmospheric scattering is not considered here, since all ALS to PAD methods rely on the reflection of the lidar pulse from solid elements, which are much larger than scattering off of aersols.

We suggest the updated sentence: *The reason for only using one-way attenuation is that the scattering from canopy elements and ground does not come from continuous attenuation, but rather distinct reflection on objects.*

8. *P6 L19 typo on 'therefore'*

Updated.

9. *Figure 2 legend: the bright green text (and figure symbols) is not easy on the eyes.*

The colors were chosen to give a good representation for people with normal vision as well as color blind readers and have been carefully considered. We have enlarged the scatter plots and included confidence bounds in colors, so the markers are now black. It is our opinion that the readability has improved.

10. *Figure 2 and 3: histograms would also be interesting to compare.*

Probability density functions were presented in Fig. 6 of the original manuscript We have kept them there in the revised version.

11. *P. 10 L. 22: without a 'true' value it isn't accurate to say that one method or another is an 'overestimation'. This could however be determined in a simulation.*

We realize it is not technically correct to determine whether or not there is an overestimation by the FR or AR methods. However, as can be seen in Fig. 6 and Table 2, both the median and the mean value of the PAI in the winter scan is high for FR and AR relative to SR and IR. In comparison to published values of winter PAI for beech forests, both are above the maximum finding in Bréda (2003). In addition to that, the ratio of winter to summer PAI is in excess of 0.4 for both FR and AR, also that above the maximum presented in Bréda (2003). Here the authors consider that there is a clear theoretical limitation of the FR method that the intensity of the reflection is not taken into account. As seen in Fig. 1, the canopy in wintertime, while clearly sparser than in summer, still has few large gaps. The beam width for the winter scan of Falster was around 40 cm, larger than many of the gaps seen in Fig. 1 b.

We suggest the updated sentence:

"The reason for the high PAI values in FR and AR relative to SR and IR in the case of Falster winter becomes apparent from Fig. 5. The ground returns are biased towards higher order returns, which are not included in FR. As seen in Fig. 1 b, many of the gaps in the winter canopy are smaller than the  40 cm beam width for the Falster winter scan, making first order returns in the lower canopy less likely. AR, on the other hand, include the higher order returns, but neither FR nor AR takes into account the low intensity in the returns from the upper canopy, ultimately leading to a relatively higher estimate of PAD and PAI."

Furthermore we have added in the discussion, after l.11, p.15:

"The winter estimates of average PAI in FR and AR exceed the maximum value for Beech trees reported in Bréda (2003). In addition to that, the ratio of winter to summer PAI is in excess of 0.4 for both FR and AR, also that above the maximum presented in Bréda (2003)."

12. *p. 15: remove 'dubious'.*

We have replaced it with the word doubtful. Furthermore we have added a clarification that the AR method was only included in the complementary material of the Almeida et al. (2019) study, partly due to the unknown impact of the violation of the assumption that all returns are independent. We still think that the AR method is interesting to include, partly because Almeida et al. (2019) omitted the effect of the beam inclination angle (included in this study), and partly to show how it compares to the other methods.

The comments from the reviewer have certainly helped us to improve our manuscript and we hope that the comments have been taken into consideration satisfactorily.

**References**

Almeida, D. R. A. d., Stark, S. C., Shao, G., Schietti, J., Nelson, B. W., Silva, C. A., Gorgens, E. B., Valbuena, R., Papa, D. d. A., and Brancalion, P. H. S.: Optimizing the Remote Detection of Tropical Rainforest Structure with Airborne Lidar: Leaf Area Profile Sensitivity to Pulse Density and Spatial Sampling, Remote Sensing, 11, 92, https://doi.org/10.3390/rs11010092, 2019.

5 Boudreault, L. É., Bechmann, A., Tarvainen, L., Klemedtsson, L., Shendryk, I., and Dellwik, E.: A LiDAR method of canopy structure retrieval for wind modeling of heterogeneous forests, Agricultural and Forest Meteorology, 201, 86–97, https://doi.org/10.1016/j.agrformet.2014.10.014, 2015.

Bréda, N. J. J.: Ground based measurements of leaf area index: a review of methods, instruments and current controversies, Journal of Experimental Botany, 54, 2403–2417, https://doi.org/10.1093/jxb/erg263, https://dx.doi.org/10.1093/jxb/erg263, 2003.

10 Freier, J.: Approaches to characterise forest structures for wind resource assessment using airborne laser scan data, Master's thesis, University of Kassel, Kassel, Germany, http://publica.fraunhofer.de/dokumente/N-464768.html, 2017.

Hopkinson, C. and Chasmer, L.: Remote Sensing of Environment Testing LiDAR models of fractional cover across multiple forest ecozones, Remote Sensing of Environment, 113, 275–288, https://doi.org/10.1016/j.rse.2008.09.012, http://dx.doi.org/10.1016/j.rse.2008.09.012, 2009.

15 Morsdorf, F., Kötz, B., Meier, E., Itten, K. I., and Allgöwer, B.: Estimation of LAI and fractional cover from small footprint airborne laser scanning data based on gap fraction, 104, 50–61, https://doi.org/10.1016/j.rse.2006.04.019, 2006.

Richardson, J. J., Moskal, L. M., and Kim, S.-H.: Modeling approaches to estimate effective leaf area index from aerial discrete-return LIDAR, Agricultural and Forest Meteorology, 149, 1152 – 1160, https://doi.org/https://doi.org/10.1016/j.agrformet.2009.02.007, http://www.sciencedirect.com/science/article/pii/S0168192309000409, 2009.

20 Solberg, S., Næsset, E., Hanssen, K. H., and Christiansen, E.: Mapping defoliation during a severe insect attack on Scots pine using airborne laser scanning, Remote Sensing of Environment, 102, 364–376, https://doi.org/10.1016/j.rse.2006.03.001, http://linkinghub.elsevier.com/retrieve/pii/S0034425706000964, 2006.

---

## Author Response (AR1)

**Cover letter**

**Robust processing of airborne laser scans to plant area density profiles**

Johan Arnqvist1, Julia Freier2, and Ebba Dellwik3 Johan Arnqvist, Uppsala University, department of Earth Sciences, Uppsala, Sweden 2Fraunhofer Institute for Energy Economics and Energy System Technology, Kassel, Germany 3Technical University of Denmark, Roskilde, Denmark Correspondence: Johan Arnqvist (johan.arnqvist@geo.uu.se)

For the version of the manuscript, we have made major updates and changes. Most of these changes are described and argued for in the answers to the referees following their first assessment. Compared to the original manuscript, the changes are marked in red in the following manuscript, and are also briefly outlined below.

**Short outline of the major changes**

- We have created a subsection of the method section that describes the evaluation parameters that we have used to compare the different routines to each other and to ground based methods. The results section has been reworked to follow the section describing the evaluation parameters.
- As one of the evaluation parameters we devised a new test for the sensitivity of the methods to the ground albedo.
- The background and introduction have been reworked to clarify the purpose and aim of the study. More reference literature has been included.
- The description of the methods to derive plant area density from airborne laser scans has been updated to better highlight their foundation and differences
- The description of the new method has been reworked in order to better explain how it combines the benefits of the earlier methods
- Information regarding scanning characteristics has been added in a new table. This was deemed important to explain how the relation between footprint size of the laser beam and the forest stand characteristics influences the PAD/PAI estimation methods.
- The map figures have been updated to better show differences between the routines, and regression lines as well as confidence bounds have been added to the comparison with the ground based routines
- Both the discussion and conclusion have been updated and shortened to better reflect the main outcomes of the study

The authors are grateful to the two anonymous referees and it is our judgement and hope that this version of the manuscript is an improvement compared to the first version.

[revised manuscript text omitted]